# Targeted Metabolomics to Assess Exposure to Environmental Chemicals of Concern in Japanese Quail at Two Life Stages

**DOI:** 10.3390/metabo11120850

**Published:** 2021-12-08

**Authors:** Elena Legrand, Niladri Basu, Markus Hecker, Doug Crump, Jianguo Xia, Bharat Chandramouli, Heather Butler, Jessica A. Head

**Affiliations:** 1Faculty of Agricultural and Environmental Sciences, McGill University, Montréal, QC H9X 3V9, Canada; niladri.basu@mcgill.ca (N.B.); jeff.xia@mcgill.ca (J.X.); jessica.head@mcgill.ca (J.A.H.); 2Toxicology Centre and School of the Environment and Sustainability, University of Saskatchewan, Saskatoon, SK S7N 5B3, Canada; markus.hecker@usask.ca; 3Environment and Climate Change Canada, National Wildlife Research Centre, Carleton University, Ottawa, ON K1S 5B6, Canada; doug.crump@ec.gc.ca; 4SGS-AXYS Analytical Services Ltd., 2045 Mills Road West, Sidney, BC V8L 5X2, Canada; Bharat.Chandramouli@sgs.com (B.C.); hjbutler@blackbeckresearch.com (H.B.)

**Keywords:** environmental metabolomics, avian, early life stage, mass spectrometry, LC-MS

## Abstract

This proof-of-concept study characterizes the Japanese quail (*Coturnix japonica*) hepatic metabolome following exposure to benzo[a]pyrene, chlorpyrifos, ethinylestradiol, fluoxetine hydrochloride, hexabromocyclododecane, lead(II)nitrate, seleno-L-methionine, and trenbolone in embryos and adults. The analysis revealed effects on lipid metabolism following exposure to several chemicals at both life stages. The most pronounced effects were observed in embryos exposed to 41.1 μg/g chlorpyrifos. This work highlighted challenges and the need for further avian metabolomics studies.

## 1. Introduction

Metabolomics in ecotoxicology generates high throughput data, providing information on alterations of specific metabolic pathways in response to environmental stress, including anthropogenic contaminants [1]. Metabolite changes have traditionally been primarily measured using nuclear magnetic resonance spectroscopy (NMR), but liquid chromatography, coupled with mass spectrometry (LC-MS)-based metabolomics, is now increasingly used, due to its higher sensitivity [2].

Although the number of metabolomics studies in ecotoxicology has drastically increased over the past 20 years, targeted and untargeted LC-MS-based metabolomics (in support of ecotoxicological studies) have primarily focused on assessing metabolite alterations in aquatic organisms exposed to a variety of chemicals [1,2]. For example, a targeted analysis reported that arginine, proline, alanine, aspartic acid, and glutamate metabolism were altered after PCB exposure in both zebrafish (*Danio rerio*) embryos and larvae [3]. An untargeted analysis revealed that the UV filter oxybenzone affected lipid metabolism, and phenylalanine and tyrosine metabolism in the liver and plasma of the gilt-head bream (*Sparus aurata*) [4].

In contrast, metabolomics studies have rarely been applied to the field of avian ecotoxicology, and have only focused on one of two species: chicken or double-crested cormorant [5,6,7,8,9,10]. These studies revealed that lipid metabolism was affected by exposure to perfluorooctanesulfonic acid, chromium, and alpha-cypermethrin in chickens [6,7,10]. In addition, amino acid, energy, fatty acid, and nucleoside metabolisms were disrupted in double-crested cormorant and chicken after exposure to Deep Water Horizon oil and T2-toxin, respectively [8,9]. Recently, a metabolomics study in Japanese quail embryos revealed effects on lipid and fatty-acid metabolism following chlorpyrifos exposure [11].

Here, we describe a proof-of-concept metabolomic study in the Japanese quail (*Coturnix japonica*) after exposure to anthropogenic contaminants at two life stages. Targeted LC-MS-based metabolomics data were collected as part of a large-scale Genome Canada project (www.ecotoxchip.ca, accessed on 6 December 2021) aimed at validating early-life stage (ELS) models for toxicity testing, and integrating omics approaches into screening and prioritization of environmental chemicals [12]. The EcoToxChip project assessed hepatic transcriptomics, proteomics, and metabolomics, as well as apical outcomes, histology, and chemical analysis data for eight chemicals of environmental concern, and controls in ELS and adult Japanese quail [13,14]. The objective of the present study was to investigate the effects of these eight chemicals on the Japanese quail liver metabolome.

## 2. Results and Discussion

The data presented in this study illustrate a characterization of the Japanese quail metabolome at two life stages following exposure to eight environmental chemicals (Figure 1). Overall, few effects of chemical exposure on the metabolome were observed at the doses that we tested. This was notable, given that significant changes were reported at the transcriptional and organismal level in response to several of the chemicals [11,13]. Here, we discuss these results and factors that may have contributed to this outcome.

Among the chemicals tested, the high dose of CPF induced the most changes in the ELS liver metabolome. The PCA scores plot showed a clear effect of CPF high dose on the liver metabolome (81.9% of total explained variance by PC1 and PC2, Figure 2A). The high dose of CPF significantly impacted the highest number of metabolites in ELS livers with 59 metabolites displaying a statistically different concentration from the control group (one-way ANOVA, FDR < 0.05, Appendix A). Phospholipids and acylcarnitines were principally dysregulated, representing 47 of the 59 metabolites. Among them, 16 lipids showed a log2FC > |1.5| (Table 1). We previously explored this result in a related study comparing transcriptomic and metabolomic responses to CPF in two avian species: Japanese quail and double-crested cormorant (*Phalacrocorax auritus*) [11]. We found that CPF had a high impact in Japanese quail ELS liver, disrupting hepatic metabolism, causing oxidative stress, and endocrine disruption (steroid and non-steroid hormones). In addition to the alteration of the phospholipids and acylcarnitine, key genes involved in lipid and fatty-acid metabolism were dysregulated, showing a consistent response across metabolomic, transcriptomic, and organismal scales. These results are outside of CPF’s neurological effects, which is to be expected when investigating the avian liver.

The early-life stage of the Japanese quail liver metabolome was relatively unaffected by most of the remaining seven chemicals at the test concentrations. Individuals exposed to the TB medium dose, FLX low dose, BaP medium dose, and EE2 high dose separated from the control individuals on the PCA scores plot (Figure 2B–E), while HBCD, Pb, and SeMe showed no effect on the ELS liver metabolome, based on the PCA analysis (Figure 2F–H). The univariate statistical analysis identified changes in metabolite concentrations, following EE2 and FLX exposure. Phospholipids and acylcarnitines were the only metabolites that were significantly affected for both chemicals (Appendix A). Similar to CPF, the primary target tissue for FLX is the brain. However, effects of FLX on fatty-acid and lipid metabolism have been previously reported. In particular, an increase in carnitine concentration has been measured in rat plasma treated with fluoxetine [15]. As many aspects of lipid synthesis and transport are regulated by the endocrine system, a disruption of lipid metabolism is also expected following EE2 exposure [16]. While the traditional biomarker vitellogenin was not measured in the present study, we did observe variation in concentrations of several phosphatidylcholines in the embryonic livers. Similar results were observed in fathead minnow exposed to EE2 [16].

As with the ELS quail, few chemical-associated effects were observed in the adult Japanese quail liver metabolome. The PCA scores plot did not show a clear effect of any chemicals at any of the doses tested (Appendix A). Very few changes in metabolite concentrations were identified by one-way ANOVA (Appendix A). TB induced the most variation in the adult Japanese quail liver metabolome, with five metabolite concentrations being statistically different between treated and control individuals (FDR < 0.05, Appendix A). Two amino acids and biogenic amines (asymmetric dimethylarginine and phenylalanine) were upregulated, while three metabolites associated with energy pathways (reduced and oxidized glutathione and pentose-phosphate) were downregulated, following exposure to all three TB doses (Appendix A). Asymmetric dimethylarginine and pentose-phosphate demonstrated the biggest changes following TB in the adult liver (Table 1). The elevated concentration of asymmetric dimethylarginine could have resulted from an increase of low-density lipoprotein cholesterol due to TB exposure [17,18]. In addition, the observed glutathione depletion in adult Japanese quail liver samples could have been a function of TB metabolism [13]. Depletion of glutathione is a biomarker of oxidative stress, which has been observed in rat hepatic cells after exposure to anabolic–androgenic steroids [19]. While oxidative stress is not the primary mode of action of TB, it has been reported that TB can be detoxified by conjugation to glutathione [20].

The lack of a metabolite response to many of the chemicals should be interpreted with caution. One hypothesis could be that the concentrations tested did not alter the Japanese quail liver metabolome. As reported in Farhat et al. [13] and Boulanger et al. [14] for ELS and adult Japanese quail, respectively, the administered concentrations of Pb and SeMe were low. No apical outcomes and low-to-no transcriptomic responses were observed for these two chemicals and concentrations and, therefore, limited effects in the metabolome were expected. However, we previously showed that the concentrations of BaP, EE2, FLX, HBCD, and TB induced apical outcomes [13] and/or moderate-to-elevated transcriptomic response in Japanese quail embryos (unpublished in-house data; intended for publication elsewhere). This suggests that effects may have occurred in the metabolome that we were not able to detect with the present study design.

Our study used a targeted metabolomics approach, where 234 metabolites were investigated (Appendix A). While this technique permits the measurement of specific metabolite classes (i.e., well-known metabolites or potential-known biomarkers of environmental stress), it does not permit the discovery of novel metabolites. Since little is known about the Japanese quail liver metabolome, some chemicals and doses could potentially have impacted metabolites outside of the targeted 234. Additionally, the sample size was low for this study, which limited the statistical power and, therefore, the detection of effects.

Despite an overall low effect on the metabolome, some common effects between chemicals were observed. Phospholipids and acylcarnitines were commonly affected by three chemicals (CPF, EE2, and FLX, Table 1 and Appendix A) in ELS liver samples, and by two chemicals in adult liver samples (FLX and SeMe, Appendix A). In Japanese quail, a change in acylcarnitine concentrations could illustrate a disruption of hepatic lipid β-oxidation, which could negatively affect chick growth [21,22]. Moreover, five phospholipids (PC aa 36:1, PC aa 34:1, PC aa 34:2, PC ae 36:2, and PC aa 36:2) were impacted by both CPF and EE2 in ELS Japanese quail liver, and PC ae C38:6 was commonly affected by FLX and SeMe in adult Japanese quail (Table 2). A change in phospholipids could affect the cellular membranes and overall molecular activity [23]. The avian liver plays a primary role in lipid synthesis [24], and lipid metabolism is often affected by chemical exposure in avian species. For example, perfluorooctanesulfonic acid exposure affected lipid concentrations in the chicken embryo [10]. Moreover, transcriptomics and metabolomics analyses revealed an effect on lipid metabolism in double-crested cormorant and chicken after exposure to oil, triclosan, perfluorooctane sulfonate, and tris(1,3-dichloro-2-propyl) phosphate [9,25,26,27].

While ELS tests are increasingly being used in ecotoxicology, their application in avian metabolomics remains challenging. In the present study, the metabolomics analysis was performed on 50 mg and 100 mg (wet weight), respectively, for ELS and adult Japanese quail. To meet this weight requirement, 3 to 4 livers were pooled per sample in ELS Japanese quail, increasing the number of organisms required for the experiment. When designing metabolomics studies in ecotoxicology, biological replicates are primordial to perform powerful statistical analysis and to identify metabolite changes [28]. Due to the given constraints of the project, three pooled replicates for ELS and five individual replicates for adults were used. The low sample size of the present study limited the statistical power of the analysis and, therefore, the detection of effects. Saccenti and Timmerman [29] demonstrated that with a larger sample size, the variability of PCA loadings decreased (for a sample size greater than 50), and the stability or estimated PCA loadings increased (for a sample size greater than 25). However, this sample size seems unrealistic and unattainable for the majority of environmental metabolomics studies.

Tools have been developed to help determine the appropriate sample size for metabolomics studies. Based on a pilot study, the Power Analysis module of MetaboAnalyst can compute a sample size for the desired FDR [30]. Such analysis can help researchers to determine the appropriate number of biological replicates, balancing statistical power and experiment constraints. Based on the present CPF dataset in ELS Japanese quail, increasing the number of biological replicates to 9 to 15 would give a predicted power of 0.7 to 0.8, for a 0.05 FDR. We hope that this result will be helpful for future avian metabolomics studies.

## 3. Materials and Methods

### 3.1. Chemicals and Working Solutions

Stock solutions of eight chemicals—benzo[a]pyrene (BaP), chlorpyrifos (CPF), ethinylestradiol (EE2), fluoxetine hydrochloride (FLX), hexabromocyclododecane (HBCD), lead(II)nitrate (Pb), seleno-L-methionine (SeMe), and trenbolone (TB)—were prepared in DMSO for the ELS tests and in corn oil for the adult tests, as previously described [13,14].

For both life stages, birds were exposed to three concentrations of test chemicals and a solvent control (Figure 1). The high dose was selected to cause ≤20% mortality based on a review of the literature. The medium dose and low dose were 10- and 100-fold dilutions of the high dose, respectively (Table 3).

### 3.2. Egg Injection and Tissue Collection

A detailed description of the methodology for the ELS experiments is provided elsewhere [13]. Briefly, the eggs (n = 35) were injected on embryonic day 0 (ED0), prior to incubation (Figure 1). On ED9, a subset of embryos (n = 20) was euthanized, and livers were collected for omics analysis. To obtain sufficient wet weight required for metabolomics analysis, livers from three to four individuals were pooled, constituting one replicate, and three replicates were collected per dose. Additional liver samples were collected for other analyses, such as analytical determination of the concentrations of test chemicals in tissue, transcriptomics, proteomics, and histology [11,13]. Tissue samples were flash frozen on dry ice and stored at −80 °C until further analysis.

### 3.3. Adult Exposure and Tissue Collection

The adult exposures were carried out at EAG Agroscience, LLC facility. Hatched Japanese quail (1- or 2-day old; Loudounberry Farm & Garden, Leesburg, VA 20176, USA) were raised for 7 to 12 weeks prior to exposure, as previously described [14]. A single dose of each test chemical in corn oil was orally administered to individual Japanese quail by gavage at the start of the experiment (Figure 1). The dose volume corresponded to 4 mL/kg body weight. Four days after the exposure, subsets of 6 birds (3 per sex) per chemical and dosage group were euthanized using CO_2_ gas, and individual livers were collected and weighed. Five livers (from 3 males and 2 females) were subsectioned to yield sufficient tissue for metabolomics, transcriptomics, and proteomics, and the remaining liver was used for analytical determination of the concentrations of test chemicals in the tissue. Tissue samples were flash frozen on dry ice and stored at −80 °C until further analysis.

### 3.4. Targeted Metabolomics

#### 3.4.1. Sample Processing

The metabolomics samples were extracted at SGS AXYS Analytical Services Ltd. (Sidney, BC, Canada). Metabolite extraction was performed on approximately 50 mg of tissue per sample for ELS and 100 mg for adults, using 3 sequential methanol extractions in a bead blender. Portions of extracts (20 µL for amino acids and biogenic amines, 50 µL for fatty acids, hexose, bile acids, phospholipids, and acylcarnitines, and 250 µL for metabolites associated with energy pathways) were plated on 96-well plates preloaded with internal standards specific to the analysis. The plates were dried under liquid nitrogen. Extracts used for the analysis of amino acids and biogenic amines were derivatized using Edman’s Reagent to form phenylthiocarbamyl derivatives and then re-dried. Dried samples were resuspended in 250 µL of 5 mM ammonium acetate in MeOH for the analysis of amino acids, biogenic amines, fatty acids, hexose, bile acids, phospholipids, and acylcarnitines, and in 200 µL of MeOH for the analysis of metabolites associated with energy pathways. The plates were shaken for 20 min at 22 rpm and samples were eluted by centrifugation (100× *g* for 2 min). Samples were diluted with an equal volume of water for the analysis of amino acids, biogenic amines, phospholipids, and acylcarnitines. Samples were diluted with an equal volume of 5 mM ammonium acetate in MeOH for the analysis of fatty acids, hexose, and bile acids, and samples were diluted using 300 µL of acetonitrile for the analysis of metabolites associated with energy pathways.

The 211 samples were processed in 3 batches of 81, 80, and 50 samples. Samples were segregated based on their developmental stage (ELS versus adult), and chemical doses and associated solvent controls. The plating order within each batch was randomized across the samples. Each batch of samples was processed along with 3 procedural blank samples and 3 internal reference materials (IRMs). This allowed the extraction efficiency and reproducibility to be assessed. Metabolite concentration values in the procedural blanks represented background levels.

#### 3.4.2. Mass Spectrometry

The metabolomics samples were processed at SGS AXYS Analytical Services Ltd. (Sidney, BC, Canada). A total of 234 metabolites (Appendix A) grouped into the following classes were investigated: amino acids and biogenic amines (43 metabolites), fatty acids (17 metabolites), hexose (1 metabolite), bile acids (13 metabolites), phospholipids, and acylcarnitines (144 metabolites), and metabolites associated with energy pathways (16 metabolites). Amino acids, biogenic amines, fatty acids, hexose, bile acids, and metabolites associated with energy pathways were analyzed by LC-MS/MS, using an Agilent 1100 HPLC coupled to an API4000 triple quadrupole mass spectrometer (Applied Biosystems, Concord, ON, Canada). Phospholipid and acylcarnitine analytes were analyzed using flow-injection tandem mass spectrometry (FI-MS/MS). Mass spectrometry performance across the run was monitored by replicate injections of a mid-point calibration sample approximately once every 20 samples.

#### 3.4.3. Quantification

All data were initially processed using AB/Sciex’s Analyst Software version 1.6.2. For the quantification of phospholipids and acylcarnitines, area counts from the Analyst Software were transferred to the SGS AXYS laboratory information system (LIMS), where they were processed using an isotopic correction algorithm [31] that accounted for carbon isotope contributions between the high abundance phosphatidylcholines and the lower abundance sphingomyelins. All other methods used a 5–8 point quadratic calibration curve, generated using standards for both the target metabolites and isotopically labelled surrogate standards at known concentrations.

#### 3.4.4. Quality Control

Each batch was filtered according to its respective control checks. For each batch, metabolite concentrations outside of the expected range (50–150% of the targeted range) were flagged using the IRM measurements. Metabolites detected in two or three blank samples at a concentration above 33% of the median IRM value, and in more than half of the experimental samples, were excluded. When metabolites were detected in one blank sample at a concentration above 33% of the median IRM value, the blank average concentration was subtracted from the experimental concentration. Metabolites presenting variability between the IRMs (RSD > 30%) were excluded from the analysis.

### 3.5. Data Visualization and Statistical Analysis

Each dataset was normalized with its respective IRM (per LC-MS run) and Pareto scaled using MetaboAnalyst (v5) [30]. Metabolomics data were visualized by principal component analysis (PCA) (ELS: n = 3; adult n = 5 per dose per chemical). Metabolites that were significantly different between a given dose of each chemical and its respective control group were identified using analysis of variance (one-way ANOVA, FDR < 0.05 and post hoc Tukey HSD). Fold change analysis was performed between treated and control groups.

## 4. Conclusions

This work represents a proof-of-concept avian metabolomics study and a first step towards characterizing the Japanese quail liver metabolome after exposure to environmental chemicals of concern. While only the high dose of CPF had an important impact on the ELS Japanese quail liver metabolome, we highlighted a pattern of the lipid metabolism pathway being commonly impacted by several chemicals and at both life stages. Considering the important role of lipid metabolism in avian species, more research is needed to understand the consequences of its disruption by chemicals. Moreover, this work highlighted some challenges, which we hope will be helpful for future environmental metabolomics studies in birds.

## Figures and Tables

**Figure 1 metabolites-11-00850-f001:**
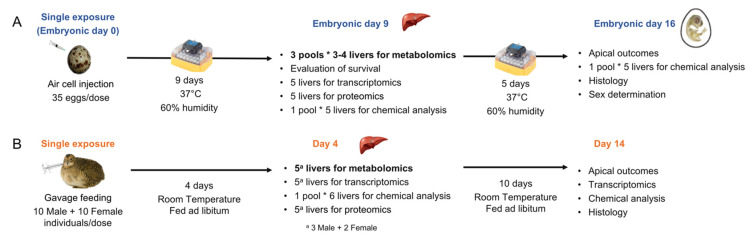
Experimental design of the early-life stage (ELS) and adult Japanese quail exposures, adapted from Farhat et al. [13] and Boulanger et al. [14]. ELS and adult Japanese quail were exposed to 8 environmental chemicals (benzo[a]pyrene (BaP), chlorpyrifos (CPF), ethinylestradiol (EE2), fluoxetine hydrochloride (FLX), hexabromocyclododecane (HBCD), lead(II)nitrate (Pb), seleno-L-methionine (SeMe), and trenbolone (TB)) following this experimental design. (**A**) Embryos were exposed via egg injection on day 0 of incubation. A subset was euthanized on embryonic day 9 and liver samples were taken for omics and chemical residue analysis. To provide sufficient tissue for metabolomics, 3–4 ELS livers were pooled for each replicate (n = 3 replicates). Remaining embryos were incubated until embryonic day 16 in order to monitor apical outcomes. (**B**) Adults were exposed to a single dose via gavage. A subset was euthanized 4 days later for omics (n = 5) and chemical residue analysis. Remaining individuals were maintained until day 14 in order to monitor apical outcomes.

**Figure 2 metabolites-11-00850-f002:**
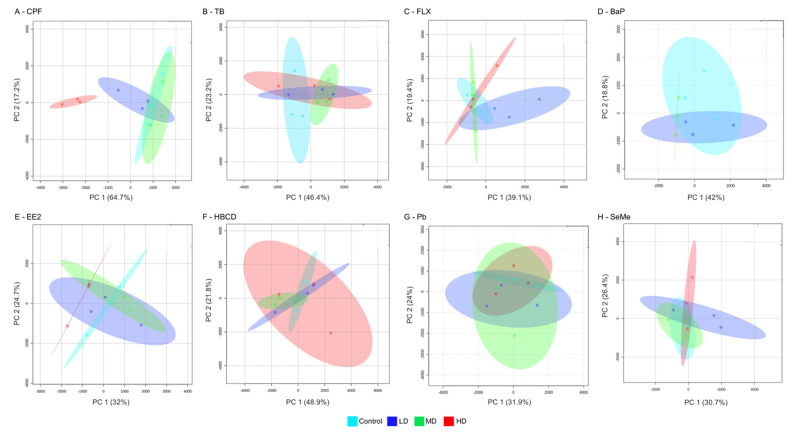
Principal component analysis (PCA) scores plot of the early-life stage (ELS) Japanese quail (JQ) liver metabolome after exposure to chlorpyrifos (CPF, **A**), trenbolone (TB, **B**), fluoxetine (FLX, **C**), benzo(a)pyrene (BaP, **D**), ethinylestradiol (EE2, **E**), hexabromocyclodecane (HBCD, **F**), lead(II)nitrate (Pb, **G**), and seleno-L-methionine (SeMe, **H**). JQ were exposed to three doses defined as low dose (LD, dark blue), medium dose (MD, green), and high dose (HD, red).

**Table 1 metabolites-11-00850-t001:** Most-impacted (log2Fold Change > |1.5|) metabolites by chemical treatment in early-life stage and adult Japanese quail liver. Statistically significant metabolites were identified by one-way ANOVA and Tukey post hoc (false discovery rate FDR < 0.05). The full list of metabolites statistically different between treated and control groups is reported in Appendix A. For better visibility, some metabolites were abbreviated: phosphatidylcholine diacyl (PC aa) and phosphatidylcholine acyl-alkyl (PC ae).

Life Stage	Chemical	Doses	Metabolite Class	Metabolite	*p*-Value	FDR	Log2(FC)		
							HD	MD	LD
ELS	CPF	HD	ABA	Ornithine	1.00 × 10^−2^	2.87 × 10^−2^	−1.62		
		HD	FHB	Arachidonic acid	7.54 × 10^−3^	2.22 × 10^−2^	−1.82		
		HD		Docosahexaenoic acid	2.07 × 10^−2^	4.65 × 10^−2^	−3.45		
		HD		FA C22:5n6c	1.11 × 10^−2^	2.96 × 10^−2^	−4.64		
		MD	LIP	AC C14	1.04 × 10^−2^	2.87 × 10^−2^		3.25	
		MD		AC C16	4.99 × 10^−3^	1.70 × 10^−2^		1.90	
		MD		AC C18:1	3.97 × 10^−4^	4.32 × 10^−3^		2.01	
		MD		AC C18:2	3.44 × 10^−5^	1.14 × 10^−3^		2.95	
		HD		lysoPC a C18:1	2.04 × 10^−3^	1.06 × 10^−2^	1.63		
		HD		lysoPC a C18:2	5.92 × 10^−3^	1.87 × 10^−2^	1.60		
		HD		PC aa C32:2	3.36 × 10^−4^	4.29 × 10^−3^	1.76		
		HD		PC aa C34:1	2.07 × 10^−4^	3.01 × 10^−3^	1.56		
		HD		PC aa C34:2	8.09 × 10^−4^	6.71 × 10^−3^	1.71		
		HD		PC aa C34:3	2.76 × 10^−3^	1.17 × 10^−2^	1.82		
		HD		PC aa C36:1	1.16 × 10^−5^	4.83 × 10^−4^	1.85		
		HD		PC aa C36:2	7.35 × 10^−5^	2.03 × 10^−3^	1.90		
		HD		PC aa C36:3	5.05 × 10^−4^	4.65 × 10^−3^	1.62		
		HD		PC ae C42:2	1.59 × 10^−3^	9.25 × 10^−3^	2.04		
		HD		PC ae C42:4	4.16 × 10^−4^	4.32 × 10^−3^	−2.04		
		HD;MD;LD		SM C26:1	1.89 × 10^−3^	1.01 × 10^−2^	−3.07	−2.27	−2.73
	EE2	HD;MD;LD	LIP	PC aa C40:2	2.23 × 10^−7^	3.71 × 10^−5^	−2.60	−2.34	−2.74
Adult	TB	HD; MD; LD	ABA	Asymmetric dimethylarginine	6.99 × 10^−5^	5.15 × 10^−3^	1.66	1.95	1.90
				Pentose- phosphate	6.10 × 10^−4^	2.69 × 10^−2^	−2.43	−2.83	−2.62

Chemicals are defined as follows: chlorpyrifos (CPF), ethinylestradiol (EE2), and trenbolone (TB), and the doses as high dose (HD), medium dose (MD), and low dose (LD).

**Table 2 metabolites-11-00850-t002:** Metabolites impacted by multiple chemicals in early-life stage (ELS) and adult quail. None of the metabolites were impacted at both life stages. Chemicals are abbreviated as chlorpyrifos (CPF), ethinylestradiol (EE2), trenbolone (TB), and seleno-L-methionine (SeMe).

Metabolite	Chemicals	Life Stage
PC ae C36:2	CPF, EE2	ELS
PC aa C34:1	CPF, EE2	ELS
PC aa C34:2	CPF, EE2	ELS
PC aa C36:1	CPF, EE2	ELS
PC aa C36:2	CPF, EE2	ELS
PC ae C38:6	FLX, SeMe	Adult

**Table 3 metabolites-11-00850-t003:** Administered concentrations of test chemicals in early-life stage (ELS) and adult Japanese quail. As described in Figure 1, ELS Japanese quail were injected with a single chemical dose, while adult Japanese quail were exposed via gavage. Vehicle solvents were DMSO and corn oil for ELS and adult, respectively. Values in the table were calculated based on the analytically determined (ELS) or nominal (adult) concentrations of the dosing solutions. Chemicals are abbreviated as follows: ethinylestradiol (EE2), chlorpyrifos (CPF), benzo[a]pyrene (BaP), lead(II)nitrate (Pb), seleno-L-methionine (SeMe), fluoxetine hydrochloride (FLX), trenbolone (TB), and hexabromocyclododecane (HBCD). JQ were exposed to three doses of each chemical (low dose: LD, medium dose: MD, and high dose: HD). Table was adapted from Farhat et al. [13] and Boulanger et al. [14].

Chemical	ELS			Adult		
	Administered Concentration (ppm ^a^ egg)			Administered Concentration (ppm ^a^)		
	LD	MD	HD	LD	MD	HD
EE2	0.54	6.3	54.2	0.05	0.5	5
CPF	0.56	4.9	41.1	0.1	1	10
BaP	0.01	0.05 ^b^	0.83	0.5	5	50
Pb	0.07 ^c^ (0.11)	0.7 ^c^ (1.1)	6.7 (10.7)	35	350	3500
SeMe	0.0003 ^d^ (0.0007)	0.002 ^d^ (0.005)	0.03 ^d^ (0.07)	0.1	1	10
FLX	0.39	4.6	32.7	1	10	100
TB	0.04	0.43	4.4	0.1	1	10
HBCD	0.02γ	0.73γ	10.5γ	10	100	1000

^a^: Parts per million relative to egg and body weight for ELS and adult Japanese quail, respectively. ^b^: Nominal value (due to an error, this stock solution was not analyzed). ^c,d^: Reported values represent the analytically determined concentrations of elemental lead or selenium, followed by the calculated concentrations of lead(II)nitrate or seleno-L-methionine in parentheses, where applicable. γ: γ-HBCD is the predominant isomer in the technical mixture used in this study.

## Data Availability

Data supporting the results can be found in the Appendix A.

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
