# Peer review of "Targeted Metabolomics to Assess Exposure to Environmental Chemicals of Concern in Japanese Quail at Two Life Stages"

_metabolites, 2021, doi:10.3390/metabo11120850_

Round 1
Reviewer 1 Report
All my comments have been answered. The manuscript is then suited for publication.
One minor comment : In table S1 (line 1), I guess this is log2(FC) that is represented (as for table 1) and not the FC?
Reviewer 2 Report
Its better to use in the future work proper sample size
This manuscript is a resubmission of an earlier submission. The following is a list of the peer review reports and author responses from that submission.
Round 1
Reviewer 1 Report
Dear Authors
the Work consider good start for new technique in avian species, but if the author could repeat the work with larger sample size for each group (12-20 individual). also, if evaluate oxidative stress or other marker for evaluate hepatotoxicity.
Reviewer 2 Report
Overall the authors have presented their findings well but at 18 pages long is more like a research article rather than a short communication.
I have noted some parts that need addressing and I also have suggestions with regards to it being submitted as a short communication:
Table 1 is 3 pages long, consider simplifying and maybe incorporating Table 2, as the values presented are numerous repeats of the same values.
Figure 3 needs uniformity of where the control box plot is.
Is Figure 4 required? As it's only referred to as there was no differences seen.
Page 12, line 176. 'will be published elsewhere' would be better referred to as unpublished in house data.
Reviewer 3 Report
I find the subject interesting since studies dealing with avian metabolomics are rare in the literature.
However, even if it is a short communication, I found the manuscript too much focused on description of the results with limited discussion on a molecular point-of-view. Authors state that transcriptomics analyses were performed for the chemicals investigated and that changes were highlighted. Those results seem about to be published. I would recommend either that i) the authors add these results in the present manuscript to be able to discuss the metabolites for which they were expecting changes in response to chemical exposures or ii) they remove such sentences from the manuscript and discuss the results they obtained with metabolomics in line with the MoA of the substances (for example Serotonin for FLX, Acetylcholine for CPF…)
Here are some other comments:
- Reference 14 cannot be found. Authors should give more information on this reference. If it was a communication in a congress, please provide the name of congress, where and when it was held.
- Table 1 : Fold change should be reported in the table. This information is very important to quantify the extent of the changes observed
- Table S2 : I cannot see the controls for all the chemicals in the adult assay.
- The number of replicates is low and ANOVA may not be the best test to conduct. Please use a non-parametric test (Kruskal Wallis for example).
- Authors mentioned that they checked the reproducibility with the IRM. What was the acceptable threshold for RSD, lower than 30 % as recommended for metabolomics studies ? Were the results for the metabolites presenting a RSD higher than 30 % discarded from the dataset?